# Synthesis of Silver Nanoparticles Loaded onto Polymer-Inorganic Composite Materials and Their Regulated Catalytic Activity

**DOI:** 10.3390/polym11030401

**Published:** 2019-03-01

**Authors:** Sen Yan, Chunge Jiang, Jianwu Guo, Yinglan Fan, Ying Zhang

**Affiliations:** Key Laboratory of Applied Surface and Colloid Chemistry, Ministry of Education, School of Chemistry and Chemical Engineering, Shaanxi Normal University, Xi’an 710062, China; chem-ys@snnu.edu.cn (S.Y.); jiangchunge@snnu.edu.cn (C.J.); guoJW@snnu.edu.cn (J.G.); fanyinglan@stu.snnu.edu.cn (Y.F.)

**Keywords:** polymer microgels, supported catalysts, silver nanoparticles, titania, reduction of 4-nitrophenol (4-NP)

## Abstract

We present a novel approach for the preparation of polymer-TiO_2_ composite microgels. These microgels were prepared by the in situ hydrolysis and condensation of titanium tetrabutoxide (TBOT) in a mixed ethanol/acetonitrile solvent system, using poly(styrene-*co*-*N*-isopropylacrylamide)/poly(*N*-isopropylacrylamide-*co*-methacrylic acid) (P(St-NIPAM/P(NIPAM-*co*-MAA)) as the core component. Silver nanoparticles (AgNPs) were controllably loaded onto the polymer-TiO_2_ composite microgels through the reduction of an ammoniacal silver solution in ethanol catalyzed by NaOH. The results showed that the P(St-NIPAM)/P(NIPAM-*co*-MAA)-TiO_2_ (polymer-TiO_2_) organic-inorganic composite microgels were less thermally sensitive than the polymer gels themselves, owing to rigid O–Ti–O chains introduced into the three-dimensional framework of the polymer microgels. The sizes of the AgNPs and their loading amount were controlled by adjusting the initial concentration of [Ag(NH_3_)_2_]^+^. The surface plasmon resonance (SPR) band of the P(St-NIPAM)/P(NIPAM-*co*-MAA)-TiO_2_/Ag (polymer-TiO_2_/Ag) composite microgels can be tuned by changing the temperature of the environment. The catalytic activities of the polymer-TiO_2_/Ag composite microgels were investigated in the NaBH_4_ reduction of 4-nitrophenol. It was demonstrated that the organic-inorganic network chains of the polymer microgels not only favor the mass transfer of the reactant but can also modulate the catalytic activities of the AgNPs by tuning the temperature.

## 1. Introduction

The introduction of silver nanoparticles (AgNPs) has attracted significant scientific interest due to their unique optical and electronic properties [1,2,3], remarkable catalytic activities and selectivities [4,5], and significant antibacterial activities [6,7]. AgNPs are widely used in catalysis [8,9,10], surface-enhanced Raman scattering (SERS) [11], biomedicine [12,13], and environmental treatment [14,15,16,17]. Unfortunately, AgNPs are generally unstable and easily aggregated owing to their high surface energies, which limit their application. In order to overcome these shortcomings, researchers usually stabilize AgNPs onto a support material with a different framework structure; this support material not only increases the specific surface area and improves dispersibility, but it also imparts certain functionality to the AgNPs [18,19,20]. Among various support materials, inorganic materials are interesting and promising candidates due to their unique mechanical strengths and stabilities. Commonly used inorganic support materials, such as carbon, silica, alumina, and metal oxides [21,22,23,24], have already been studied extensively.

Polymeric support materials, such as block copolymers [25], dendrimers [26], latex particles [27], microgels [28,29,30], and other polymers [31,32,33], have been used to load metal nanoparticles. Among these support materials, the synthesis of metal nanoparticles supported on polymer microgels has received much attention, owing to their tunable catalytic properties of these systems [34,35,36,37]. The catalytic activities of the supported catalysts can be controlled by regulating swelling and shrinking behavior, as well as the hydrophilic/hydrophobic transitions of the polymer microgels, which is achieved by adjusting the environmental conditions (e.g., temperature, pH) [17,31,38,39]. A typical example of a thermosensitive polymer microgel that has received much attention is poly(*N*-isopropylacrylamide) (PNIPAM), which undergoes reversible hydrophilic/hydrophobic transitions in aqueous solution at a lower critical solution temperature (LCST) at around 32 °C in aqueous medium [40]. The catalytic activities of metal nanoparticles loaded onto PNIPAM-based microgels can be modulated by changing the environmental conditions at their volume transitions within the polymer networks (by swelling and shrinking) leading to the diffusion of reactants. For example, Lu and Ballauff introduced a novel carrier system composed of thermosensitive PNIPAM networks for the immobilization of metal nanoparticles, such as Ag, Pd, Au, Rh, and Pt, as well as bimetallic Au-Pt NPs [41,42]. The results revealed that the catalytic activities of metal-polymer composite particles can be adjusted in a nonlinear fashion by varying the temperature. Our group synthesized AgNPs loaded onto/into the network chains of the poly(styrene-*N*-isopropylacrylamide)/poly(*N*-isopropylacrylamide-*co*-methacrylic acid) (P(St-NIPAM)/P(NIPAM-*co*-MAA)) polymer microgels by a facile method [43]. The introduction of non-thermosensitive PMAA chains into the network was expected to limit the volume change of the shell caused by the PNIPAM chains at the LCST, thereby achieving favorable mass transfer. Tang’s group synthesized AgNPs-embedded poly(*N*-isopropylacrylamide-*co*-2-(dimethylamino)ethylmethacrylate) (P(NIPAM-*co*-DMA)) composite microgels that exhibited excellent thermo-pH dual-responsive behavior, and regulated their surface plasmon resonance (SPR) properties by varying the pH [38]. The Ag-P(NIPAM-*co*-DMA) composite microgel exhibited high catalytic reactivity for the reduction of methyl blue (MB) in aqueous solution, which was modulated by temperature. Our group proposed a synthetic approach toward organic-inorganic microgels as supports for metal nanoparticles. This type of organic-inorganic microgel not only exhibited good mechanical strength and chemical stability, but also had a good three-dimensional network structure and the environmental sensitivity of a polymer microgel [44,45]. Poly(styrene-*N*-isopropylacrylamide)/poly(*N*-isopropylacrylamide-*co*-3-methacryloxypropyltrimethoxysilane)-Ag (P(St-NIPAM)/P(NIPAM-*co*-MPTMS)-Ag), a core-shell-type microgel, was synthesized using P(St-NIPAM)/P(NIPAM-*co*-MPTMS) composite microgels as the support material, and AgNPs were synthesized in situ in a controllable manner with ethanol as the reducing agent. The results reveal that the AgNP dispersity was greatly improved by the confining effect of the organic-inorganic-microgels network. The composite microgels exhibited excellent catalytic performance and thermosensitivity for modulating the catalytic activities of the AgNPs.

In this study, we introduce a facile approach for the synthesis of organic-inorganic P(St-NIPAM)/P(NIPAM-*co*-MAA)-TiO_2_ (polymer-TiO_2_) through the hydrolysis and condensation of titanium tetrabutoxide (TBOT) catalyzed by ammonium hydroxide in a mixed ethanol/acetonitrile solvent system, using P(St-NIPAM)/P(NIPAM-*co*-MAA) core-shell structured polymer microgels as templates. AgNPs were formed in situ on the shell layers of the polymer-TiO_2_ organic-inorganic composite microgels. To evaluate the catalytic performance of the prepared polymer-TiO_2_/Ag composite microgels, the temperature dependence of the catalytic activity for the reduction of 4-nitrophenol (4-NP) was investigated.

## 2. Materials and Methods 

### 2.1. Materials

*N*-isopropylacrylamide (NIPAM) was purified by recrystallization in n-hexane and dried in a vacuum. Styrene (St) was washed by NaOH aqueous solution (5–10 wt %) and distilled under reduced pressure. Methacrylic acid (MAA) was purified through distillation under reduced pressure prior to polymerization. The initiator (ammonium persulfate, APS), the crosslinking agent (*N*,*N*′-methylenebisacrylamide, MBA), sodium hydroxide (NaOH), tetrabutyl titanate (TBOT), acetonitrile, silver nitrate (AgNO_3_), ammonia solution, sodium borohydride (NaBH_4_), 4-nitrophenol (4-NP), and absolute alcohol were all analytically pure and were used as received. All the chemicals were purchased from Sinopharm Chemical Reagent Co., Ltd of China (Shanghai, China). Water used in the experiment was redistilled water.

### 2.2. Synthesis of Polymer-TiO_2_ Organic-Inorganic Composite Microgels

Core-shell-structured P(St-NIPAM)/P(NIPAM-*co*-MAA) polymer microgels were synthesized in two steps, namely soap-free emulsion polymerization and seeded emulsion polymerization [43]. Using the P(St-NIPAM)/P(NIPAM-*co*-MAA) composite microgels as templates, amorphous TiO_2_ was prepared through the hydrolysis and condensation of TBOT by ammonium ions in a mixture of absolute ethanol and acetonitrile [46]. Typically, 0.03 g of P(St-NIPAM)/P(NIPAM-*co*-MAA) microgels (in the dry state) was dispersed in 36 mL of a 3:1 (*v*/*v*) mixture of absolute ethanol/acetonitrile with stirring for 1 h and cooling (ice-water bath). A 0.1 mL aliquot of ammonia solution was then added into the above-mentioned dispersion. A solution of TBOT (0.05, 0.075, or 0.10 mL) in 3:1 (*v*/*v*) ethanol/acetonitrile (10 mL) was added after 15 min. The hydrolysis and condensation reactions of TBOT were conducted for 2 h under stirring (300 rpm). The final products were washed several times by centrifugation with absolute ethanol (10,000 rpm for 15 min). Different amounts of TBOT were used to prepare polymer-TiO_2_ organic-inorganic composite microgels with different compositions. The product was dispersed in 60 mL of absolute ethanol and stored at room temperature.

### 2.3. Syntheses of the Polymer-TiO_2_/Ag Composite Microgels

Polymer-TiO_2_/Ag composite materials were prepared by the in situ reduction of ammoniacal silver solution, using ethanol as the reducing agent, NaOH as the catalyst, and polymer-TiO_2_ organic-inorganic composite microgels as templates. In a typical procedure, 0.075 mL of the polymer-TiO_2_ dispersion was ultrasonically dispersed and swelled in 60 mL of absolute ethanol for 3 h while immersed in an ice-water bath, after which it was transferred into a 150 mL three-necked round-bottom flask and stirred (300 rpm, 25 °C). A specific amount of fresh [Ag(NH_3_)_2_]^+^ solution (add ammonia to AgNO_3_ solution until the initial precipitate is dissolved) was injected, and 2 mL of aqueous 0.03 M NaOH was added dropwise after 1 h. The reduction reaction was maintained at room temperature for 18 h in the dark. The polymer-TiO_2_/Ag composite microgels were finally purified through repeated centrifugation, with alternate washing with water and absolute ethanol, after which they were dried in a vacuum oven at room temperature for 24 h. In this manner, brownish yellow polymer-TiO_2_/Ag composite materials were prepared by altering the initial concentration of the [Ag(NH_3_)_2_]^+^ solution (0.05 and 0.1 M). The typical procedure is illustrated in Scheme 1.

### 2.4. Catalytic Reduction of 4-NP

Briefly, 100 μL of aqueous 4-NP solution (2.0 mM) and 3.0 mL of aqueous NaBH_4_ (0.1 M) were added to a standard quartz cuvette. A 20 μL aliquot of the composite nanocatalyst dispersion (0.15 g/L) was added into the cuvette to commence the reduction of 4-NP. UV-Vis (U-3900/3900H) spectra were recorded over the 250–600 nm range every minute in order to monitor the progress of the reaction. The apparent reaction rate constants (*k*_app_) can be obtained by measuring the changes in absorption peak at 400 nm with time. The pseudo first order kinetic equation adopted to calculate the values of *k*_app_ is given as follows:ln(*A*_t_/*A*_0_) = −*k*_app_(1)
where *A*_t_ and *A*_0_ are the absorbances of 4-NP at reaction times *t* and 0, respectively [47].

### 2.5. Characterization

The morphology and size of the prepared composite materials were observed by a transmission electron microscope (TEM, JEM-2100, JEOL, Tokyo, Japan) operating at an accelerating voltage of 200 kV. The energy-dispersive X-ray spectroscopy (EDS) of the sample was examined by field-emission transmission electron microscopy (FE-TEM, FEI, Tecnai G2 F20, Hillsboro, USA). Infrared spectra were recorded on an Avatar 360 Fourier transform infrared (FT-IR) spectrometer (Nicolet, Massachusetts, USA) using the KBr pellet technique. X-ray powder diffraction (XRD) analysis was carried out using a D/Max-3C automatic X-ray diffractometer (Rigalcu, Tokyo, Japan) with Cu Kα radiation at 35 kV and 40 mA, and a scan rate of 8 (°)/s was applied to record patterns in the 2θ range of 5–80°. X-ray photoelectron spectroscopy (XPS) measurements were performed using an AXIS ULTRA instrument (Kratos Analytical Ltd., New York, USA) equipped with a monochromatic Al Kα X-ray as the excitation source. Thermo-gravimetric analyses (TGA) were determined using a Q50 analyzer (TA, Lukens Drive, USA). The measurements were conducted at a heating rate of 10 °C/min under N_2_ atmosphere. The swelling behavior of the composite microgels was measured by a laser particle analyzer (Brooke, BI-90 Plus, Ettlingen, Germany) in a temperature range of 15 to 50 °C. UV-Vis spectra were measured on a U-3900/3900H (Hitachi, Tokyo, Japan) ultraviolet-visible spectrophotometer with a temperature controller.

## 3. Results and Discussion

### 3.1. Morphologies of the Composite Microgels

The P(St-NIPAM)/P(NIPAM-*co*-MAA) polymer microgels exhibited a uniform spherical morphology that was well monodispersed. The average diameter of the microgel particles was approximately 207 nm, as shown in Figure 1.

Figure 2 displays typical TEM images of the polymer-TiO_2_ organic-inorganic composites prepared with different amounts of TBOT. Compared to the P(St-NIPAM)/P(NIPAM-*co*-MAA) polymer microgels, the composite microgels loaded with TiO_2_ present clearer and more well-defined core-shell structures. The average diameter of the polymer-TiO_2_ composite particles was about 247 nm when 0.05 mL of TBOT was added; the average diameter increased to 254 and 262 nm when the amount of TBOT was increased to 0.075 and 0.10 mL, respectively. This is ascribable to the formation of TiO_2_ into polymer chains network with higher amounts of TBOT.

The scanning transmission electron microscope (STEM) image and elemental mapping of a polymer-TiO_2_ composites clearly reveal that the Ti element was mainly distributed in the surface regions of the P(St-NIPAM)/P(NIPAM-*co*-MAA) polymer microgels, as shown in Figure 3.

Using polymer-TiO_2_ organic-inorganic composites as templates, AgNPs were loaded onto the composite microgels by the in situ reduction of [Ag(NH_3_)_2_]^+^ ions under alkaline conditions, using absolute ethanol as a solvent and reducing agent. Here, [Ag(NH_3_)_2_]^+^ was absorbed or swollen into the network structure during the ethanol reduction process due to electrostatic interactions between [Ag(NH_3_)_2_]^+^ and the COO^−^ group. TEM images of the polymer-TiO_2_/Ag composite microgels are shown in Figure 4. The images reveal that the sizes of the AgNPs and their dispersion were affected by the initial [Ag(NH_3_)_2_]^+^ concentration. With increasing amounts of [Ag(NH_3_)_2_]^+^, the average diameters of AgNPs loaded onto polymer/TiO_2_ were approximately 5.8 and 10.8 nm, respectively, which were determined by the statistical calculations based on the TEM images. Increasing amounts of [Ag(NH_3_)_2_]^+^ led to larger AgNP sizes and loading amounts. This is ascribable to the enhancement in the reduction rate brought about by the higher initial concentration of [Ag(NH_3_)_2_]^+^, leading to the formation of more silver seeds that increasingly grow on the surfaces of polymer-TiO_2_ composite microgels.

The EDS analysis of polymer-TiO_2_/Ag composite microgels is displayed in Figure 5, which clearly shows that Ag and Ti elements were mainly distributed in the surface regions of the polymer microgels. This confirms that AgNPs successfully formed on the surfaces of the polymer-TiO_2_ composite microgels.

### 3.2. Composition Analyses of the Composite Microgels

Figure 6 displays the UV-Vis spectra of the prepared composite microgels. No distinct absorptions were observed in the 250–600 nm wavelength range of the polymer-TiO_2_ organic-inorganic composite microgels, as shown in Figure 6a. In contrast, the polymer-TiO_2_/Ag composites prepared using different initial concentrations of [Ag(NH_3_)_2_]^+^ reveal absorption bands at 442 and 470 nm, respectively, which were attributed to characteristic AgNP absorptions, as shown in Figure 6b,c. The SPR of the larger AgNPs became red shifted with increasing initial concentrations of [Ag(NH_3_)_2_]^+^ [48]. Aggregation of the AgNPs loaded onto the polymer-TiO_2_ composite microgels leads to a broadening of the absorption bands. In addition, the color of the AgNPs-loaded composite dispersions changed from white to yellow, as shown in the inset of Figure 6. The UV-Vis spectral results indicate that AgNPs were formed on the polymer-TiO_2_ composite microgels.

The component of the composite microgels was characterized by FT-IR spectroscopy, as shown in Figure 7. In the FT-IR spectra of P(St-NIPAM)/P(MAA-*co*-NIPAM), the peaks at 698 and 757 cm^−1^ were assigned to the C–H out-of-plane wagging vibration of singly substituted aromatic rings, indicating that styrene was present in the prepared materials. The peaks at 1649 and 1547 cm^−1^ were assigned to the C=O stretch of the PNIPAM amide (I) and (II) groups, whereas 1709 cm^–1^ was attributed to the asymmetrical stretching vibration of the C=O group in MAA [43]. In addition, the peaks at 3309 and 3440 cm^−1^ were attributed to the N–H stretching vibration of the PNIPAM and the stretching vibration absorption of hydroxyl function groups. Compared with the FT-IR spectra of the template microgels, the peaks at 1649, 1547, and 1709 cm^−1^ of the polymer-TiO_2_ and polymer-TiO_2_/Ag composite microgels were weakened to a certain extent, which indicated that a strong interaction occurs at TiO_2_ and AgNPs with polymer microgels [49]. The peaks changing at 3440 cm^−1^ were attributed to the stretching vibration absorption of hydroxyl function groups (TiO_2_–OH bonds) [50]. FT-IR results showed that there was no characteristic vibration of O–Ti–O bonds in the range 450–610 cm^−1^, and TiO_2_ does not form a complete shell onto P(St-NIPAM)/P(NIPAM-*co*-MAA) microgels. Thus, FT-IR results shows that the formation TiO_2_ is mainly present in the polymer chains network.

The compositions of the various materials were further analyzed by XPS. Compared to P(St-NIPAM)/P(NIPAM-*co*-MAA), as shown in Figure 8, polymer-TiO_2_ organic-inorganic composite microgels exhibited an additional peak at around 458.9 eV that is consistent with Ti 2p binding energies [51], in addition to peaks corresponding to C, N, and O. Furthermore, polymer-TiO_2_/Ag revealed a further peak at 375.1 eV, which was assigned to the Ag 3d level [8].

Figure 9 displays the Ti 2p XPS spectra of the polymer-TiO_2_ organic-inorganic composites, with the peaks observed at 457.4 and 463.2 eV corresponding to the Ti 2p 3/2 and Ti 2p 1/2 levels [50], respectively. The peaks observed at 368.3 and 374.3 eV in the spectra of the polymer-TiO_2_/Ag composite materials correspond to Ag 3d 5/2 and Ag 3d 3/2 levels, respectively. The 6.0 eV difference in the binding energies of these two peaks is ascribed to the presence of metallic Ag [39]. XPS confirmed that AgNPs were loaded onto the surfaces of the polymer-TiO_2_ organic-inorganic composite microgels.

The XRD patterns of the prepared composite materials are shown in Figure 10A. The broad peak at around 21° observed in the patterns of both materials was assigned to the amorphous polymer. In the case of the polymer-TiO_2_ organic-inorganic composite materials, no diffraction peaks appeared following TiO_2_ loading, consistent with the formation of amorphous TiO_2_ [52]. The XRD patterns of the composite microgels loaded with Ag exhibited four diffraction peaks at 38.1°, 44.3°, 64.4°, and 77.5°, which are indexed to the characteristic (111), (200), (220), and (311) diffractions of Ag crystals with face-centered cubic structures [29].

The TGA traces of the composite microgels are displayed in Figure 10B. The weight losses observed below 350 °C were attributed to the evaporation of the physically adsorbed water and residual solvent from the samples. The more significant weight losses that began at 400 °C for the three samples correspond to the pyrolysis of the polymer. Compared with P(St-NIPAM)/P(NIPAM-*co*-MAA), the mass fraction of the residue following TGA were about 22.7% and 44.3% for polymer-TiO_2_ and polymer-TiO_2_/Ag, respectively.

### 3.3. Thermosensitivity of the Composite Microgels

The average diameters of the composite particles were measured by laser particle-size analysis at different temperatures (15–50 °C), and the swelling ratios of the composite microgels as functions of temperature are shown in Figure 11. The swelling ratios of the P(St-NIPAM)/P(NIPAM-*co*-MAA) microgels decreased with increasing temperature, with clear decreases observed at 32 °C, consistent with the LCST of PNIPAM (32 °C) and its state of microgels shrinkage. Polymer-TiO_2_ and polymer-TiO_2_/Ag exhibited the decreasing swelling-ratios as polymer template, indicating that the introduction of the inorganic components, namely TiO_2_ and AgNPs, inhibit shrinkage of the microgels network to some extent [45].

The effect of temperature on the SPR of polymer-TiO_2_/Ag was investigated by UV-Vis absorption spectroscopy. The SPR wavelength of AgNPs is the relevance with the surrounding microenvironment. For AgNP loading thermosensitivity polymer microgels, the SPR wavelength of AgNPs can be tuned by swelling and shrinkage of polymer with changing temperature. The SPR wavelengths increased from 442 to 455 nm, as the temperature was raised from 25 to 50 °C, as shown in Figure 12a, which is a consequence of the shrinkage of the P(St-NIPAM)/P(NIPAM-*co*-MAA) polymer chains. Figure 12b shows the values of SPR wavelength of the polymer-TiO_2_/Ag as the temperature increases from 25 to 50 °C, followed by cooling down and then increases again. The extinction spectrum shifts from 442 nm at 25 °C to 455 nm at 50 °C, respectively. After cooling down to 25 °C, the peak returns to the previous 442 nm. This change is related to the fact that the composite microgels at 25 °C are in a swelling state, while they are in a shrinking state at 50 °C. Generally, the shrinkage of polymer microgels makes the refractive index of the surrounding microenvironment increase, resulting in the absorption spectrum of AgNPs red shift [53]. Here, the reversible variation of SPR of AgNPs with changing temperature is related to the reversible change of the refractive index rather than the aggregation. Compared with other temperature-sensitive materials, the effect of temperature on the optical properties of the prepared organic-inorganic composite microgels was relatively weak, which is a consequence of the introduction of the rigid inorganic TiO_2_.

### 3.4. Catalytic Activity for the Reduction of 4-NP

It is known that 4-nitrophenol (4-NP) is a toxic and bio-refractory pollutant that has the potential to cause considerable damage to the ecosystem and human health. Many processes for the removal of 4-NP have been developed, including methods based on adsorption, microbial degradation, photocatalytic degradation, microwave-assisted catalytic oxidation, electrocoagulation, and electrochemical treatment [54,55,56]. To evaluate the catalytic performance of the polymer-TiO_2_/Ag composites, the reduction of 4-NP by NaBH_4_ was used as a model reaction since the reduction of 4-NP in the presence of supported catalysts does not exhibit side reactions or produce by-products, and progress is conveniently monitored by UV-visible spectrophotometry in mainly aqueous medium [47]. In the absence of a catalyst, the characteristic UV-Vis absorption peak of 4-NP (400 nm) did not change over 30 min, as shown in Figure 13a. After addition of polymer-TiO_2_, the characteristic UV-Vis absorption peak of 4-NP did not change over 30 min, as shown in Figure 13b, indicating that the reduction does not occur in the absence of AgNPs.

The intensity of the 400 nm peak gradually decreased during the catalytic reduction of 4-NP, and a new absorption band at 315 nm, assigned to 4-aminophenol (4-AP), appeared following the addition of a polymer-TiO_2_/Ag catalyst to the solution, as shown in Figure 14. The reaction solution changed from light yellow to colorless, indicating that the reduction was catalyzed by the prepared composite microgels. Under the same reaction conditions (25 °C, catalyst concentration of 0.1 g/L), the time required for the complete reduction of 4-NP catalyzed by the two polymer-TiO_2_/Ag composites, prepared with initial [(AgNH_3_)_2_]^+^ concentrations of 0.05 and 0.10 M, were observed to be 32 and 9 min, respectively. These results clearly show a greatly improved reaction rate at higher Ag content in the polymer-TiO_2_/Ag composite microgels.

During catalytic reduction, pseudo-first-order kinetics were observed with respect to 4-NP because the concentration of NaBH_4_ was in large excess relative to that of the reactant (4-NP). The apparent rate constant for reduction, *k*_app_, was determined from the linear relationship between ln(*A*_t_/*A*_0_) and the reaction time. The *k*_app_ values of the two polymer-TiO_2_/Ag composite microgels (with the lower and higher AgNP loadings) were 7.89 × 10^−2^ and 3.33 × 10^−1^ min^−1^, respectively. These results reveal that the catalytic activities of these composites were related to the amounts of AgNPs loaded onto the surfaces of the polymer-TiO_2_ organic-inorganic composite microgels.

To investigate the temperature sensitivities of the NaBH_4_ reduction of 4-NP in the presence of the polymer-TiO_2_/Ag composite microgels, catalytic performance at different temperatures was examined. Variations in the absorbance of 4-NP, determined by UV-Vis spectroscopy at different temperatures, are displayed in Figure 15. The results indicated that reaction times for complete reduction were 32, 22, 27, 17, and 14 min, at temperatures of 25, 30, 32, 35, and 40 °C, respectively. The reaction rate was found to increase nonlinearly with increasing temperature; this phenomenon is related to the temperature sensitivity of the supporting microgels.

The values of *k*_app_ were determined from linear plots of ln(*A*_t_/*A*_0_) against reduction time at different reaction temperatures, as shown in Figure 16a; the dependence of *k*_app_ on temperature is shown in Figure 11b. We confirmed that the relationship between reaction rate and temperature does not obey Arrhenius behavior, with changes in *k*_app_ divided into three stages. The reaction rate accelerates in the 25–30 and 32–40 °C temperature ranges, which is attributed to the direct effect of rising temperature on the reaction rate under normal conditions. On the other hand, the anomalous decrease in *k*_app_ in the 30–32 °C temperature range is attributed to the influence of the supporting thermosensitive PNIPAM chains. When the temperature is below the LCST (25–30 °C), the PNIPAM is fully swollen, which is advantageous for the mass transfer and diffusion of substrates and leads to higher reaction rates. In the 30–32 °C temperature range, which is close to its LCST, PNIPAM segments are compacted. As a consequence, the PNIPAM segments inhibit contact between the reactants and the AgNPs. The results show that the catalytic reduction of 4-NP was effectively regulated by adjusting the temperature. It should be emphasized that AgNPs were loaded onto the shells of the polymer-TiO_2_ organic-inorganic composite microgels. TiO_2_ introduced into the shell-layer network also controlled the sizes of the in situ-formed AgNPs. In addition, the AgNP distribution into the shell favors the mass transfer of reactants that significantly enhances catalytic activity, with the above mechanism schematically illustrated in Figure 16b (inset).

### 3.5. Catalytic Cycling of the Composite Materials

In order to investigate the stabilities of the polymer-TiO_2_/Ag composite materials in the catalytic process, 4-NP was catalytically reduced at 25 °C in repeat experiments using the same catalyst sample. After each catalytic cycle, the used catalyst was separated from the reaction mixture and washed alternating with double-distilled water and absolute ethanol for several times, and dried in a vacuum oven for 24 h at room temperature. The recovered particles were then used for the next run under the same conditions. The results show that while the conversion rate of the composite catalyst decreased to some extent after four reuse cycles, it remained above 80%, as shown in Figure 17. The catalytic activity and cycling performance of the composite indicate that the organic-inorganic composite polymer improves AgNP stability; the slight reduction in catalytic activity following recycling is probably the result of the blockage of some active sites in the catalyst and the leaching of active-site AgNPs into the homogeneous phase [57].

## 4. Conclusions

In this study, polymer-TiO_2_ organic-inorganic composite microgels were synthesized by the hydrolysis and condensation of TBOT in a mixed ethanol/acetonitrile solvent system using core–shell structured P(St-NIPAM)/P(NIPAM-*co*-MAA) microgels as templates. AgNPs were fabricated onto the shell layer of polymer-TiO_2_ organic-inorganic composite microgels, and the AgNP sizes and loading amounts were clearly influenced by the initial [Ag(NH_3_)_2_]^+^ concentration. The TiO_2_ introduced into the networks of the shell layer also controlled the sizes of the in situ formed AgNPs. The SPR of AgNPs was tuned by changing the environmental temperature. In addition, the AgNP distribution into the shell favored the mass transfer of reactants that significantly enhanced catalytic activity for the reduction of 4-NP. More importantly, the AgNP composite materials were easily separated and reused in subsequent reductions. The approach introduced herein is significant and may impact the synthesis of similar inorganic-organic composite catalysts capable of thermosensitive modulation.

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
