# Peer review of "Synthesis of Silver Nanoparticles Loaded onto Polymer-Inorganic Composite Materials and Their Regulated Catalytic Activity"

_polymers, 2019, doi:10.3390/polym11030401_

Round 1

Reviewer 1 Report

The manuscript “Synthesis of Silver Nanoparticles Loaded onto Polymer-Inorganic Composite Materials and Their Regulated Catalytic Activity” described the preparation, characterization and catalytic properties evaluation of novel materials based on Ag nanoparticles supported onto a core-shell TiO2 –polymer microgels.

The paper presents many interesting data, but the overall discussion of the results is very poor, with many confusions.There are many points to be explained in order to improve the work, as follows:

1)    Line 14: the authors declare the synthesis of the TiO2-polymer composite “ using (P(St-NIPAM/P(NIPAM-coMAA)) as the core component”. In others sections (Ex. Line 61) they claim that “ composite microgel composed of core-shell structured (P(St-NIPAM/P(NIPAM-coMAA)), which means that P(St-NIPAM) is the core. The text is confusing for the reader, thus the authors should explain the core-shell morphology of the composite materials in a unitary manner.

2)    Line 81 “ thermo-sensitivities” is plural. The obtained material exhibits thermal sensitivity in different ways? Also at line 279.

3)    Line 90 “stabilities of the microgels”.  Why the authors used the plural? Is discussed more than one stability (stability against more than one factor)?

4)    Line 91 “ The introduction of O-Ti-O units … into the polymer network … provides higher specific surface areas that greatly improve the catalytic performances of Ag NPs” The sentence is not supported by experimental data and it is not reasonable. Authors should add specific surface data or remove the paragraph.

5)     Line 177 “ This is ascribable to the formation of TiO2 layers “ – this paragraph about the increase of the particle size suggests that its reason is the formation of a distinct layer of TiO2 onto the polymeric microparticles. Line 184 “….reveals that Ti element is mainly distributed in the shell region of the (P(St-NIPAM/P(NIPAM-coMAA)) microgels”. The authors claims at line 89 that TiO2 is present as O-Ti-O in polymer chains network. A rigorous description of the location and morphology of TiO2 component is needed.

6)    The author should provide an explanation for the preferential location of TiO2 component in the P(NIPAM-coMAA) shell.

7)    Line 206 “Fig. 5 clearly shows that Ag and Ti elements are distributed in the shell regions ..” Ag NPs are “in the shell” or “loaded onto” polymer?

8)     Line 221 “ Aggregation of the Ag NPs loaded onto … the microgel” No data about the original size of the Ag NPs are presented, in order to claim that they are aggregate on the polymeric particles.

9)    FTIR spectra of (P(St-NIPAM/P(NIPAM-coMAA))-TiO2 should be presented and discussed in order to support the conclusions on the interaction of TiO2 with the polymeric chain (see comment on line 89). The spectrum of the polymer- TiO2 composite is missing, thus it is difficult to evaluate the interaction of the polymeric matrix with Ag NPs.

10) In order to facilitate the understanding of provided explanation of FTIR data (lines 230-236) the peaks identification in Fig 7 will be useful.

11) Line 278: The loss of mass during the pyrolysis of (P(St-NIPAM/P(NIPAM-coMAA))-TiO2 and (P(St-NIPAM/P(NIPAM-coMAA))-TiO2/Ag NPs microgels does not support the statement that “ TiO2 and Ag NPs were successfully loaded onto the surface of polymer microgels”. Besides, the TGA analysis is not a relevant method for the characterization of the core-shell morphology and nanoparticles loading in a specific location of a composite material.

12) Line 293: What is the relevance of the study of the SPR dependence with the temperature?

13) Line 375: This statement is not supported by data provided- the influence of the TiO2 presence of the Ag NPs size is not studied .

14) Line 400: “the AgNPs sizes and loading amounts were clearly 400 influenced by the initial [Ag(NH3)2]+ concentration” No data about the size of Ag NPs is shown in the paper.

15) Line 403 Why to tune the SPR?

Author Response

The authors appreciate very much for the reviewer’s valuable comments on our manuscript. All the comments have been seriously considered. And following the suggestions, we revised the manuscript with additional experimental data and related discussions provided in the revised manuscript.

Response to reviewer 1:

1)    Line 14: the authors declare the synthesis of the TiO2-polymer composite “using (P(St-NIPAM/P(NIPAM-co-MAA)) as the core component”. In others sections (Ex. Line 61) they claim that “composite microgel composed of core-shell structured (P(St-NIPAM/P(NIPAM-co-MAA)), which means that P(St-NIPAM) is the core. The text is confusing for the reader, thus the authors should explain the core-shell morphology of the composite materials in a unitary manner.

Response: We appreciate for the reviewer’s comments. In the revised manuscript, we declare the TiO2-polymer composite microgels exhibited core-shell structure using (P(St-NIPAM/P(NIPAM-co-MAA)) as the core component in a unitary manner. 

2)    Line 81 “thermo-sensitivities” is plural. The obtained material exhibits thermal sensitivity in different ways? Also at line 279.

Response: Thanks for your suggestions. In the revised manuscript, we have modified the plural “thermosensitivities” into singular “thermosensitivity” at line 81 and line 279. 

3)    Line 90 “stabilities of the microgels”.  Why the authors used the plural? Is discussed more than one stability (stability against more than one factor)?

Response: Thanks for your suggestions. In the revised manuscript, we have modified the plural “stabilities of the microgels” into singular “stability of the microgels” at line 90. 

4)    Line 91 “The introduction of O-Ti-O units … into the polymer network … provides higher specific surface areas that greatly improve the catalytic performances of Ag NPs” The sentence is not supported by experimental data and it is not reasonable. Authors should add specific surface data or remove the paragraph.

Response: Thanks for your suggestions. In the revised manuscript, we have removed this paragraph. 

5)     Line 177 “This is ascribable to the formation of TiO2 layers “ – this paragraph about the increase of the particle size suggests that its reason is the formation of a distinct layer of TiO2 onto the polymeric microparticles. Line 184 “….reveals that Ti element is mainly distributed in the shell region of the (P(St-NIPAM/P(NIPAM-co-MAA)) microgels”. The authors claims at line 89 that TiO2 is present as O-Ti-O in polymer chains network. A rigorous description of the location and morphology of TiO2 component is needed.

Response: Thanks for your suggestions. According to the thermosensitivity of the composite microgels, comparing with the P(St-NIPAM)/P(NIPAM-co-MAA) microgels, P(St-NIPAM)/P(NIPAM-co-MAA)-TiO2 exhibited the decreasing swelling-ratios, indicating that the introduction of the O-Ti-O into polymer chains network which inhibit shrinkage of the microgels network to some extent. Therefore, the formation TiO2 is mainly present as O-Ti-O into polymer chains network. In the revised manuscript, we have modified the description of the location and morphology of TiO2 component. 

6)    The author should provide an explanation for the preferential location of TiO2 component in the P(NIPAM-co-MAA) shell.

Response: Thanks for your suggestions. According to the FT-IR results of the prepared microgels, there is no characteristic vibration of O-Ti-O bonds in the range of 450-610 cm-1. Thus, TiO2 doesn’t form a complete shell onto P(St-NIPAM)/P(NIPAM-co-MAA) microgels. However, the thermosensitivity of the composite microgels exhibited the decreasing swelling-ratios, indicating that the introduction of the TiO2 into polymer chains network which inhibit shrinkage of the microgels network to some extent. Therefore, the formation TiO2 is mainly present into polymer chains network. In the revised manuscript, we have modified the description of the location of TiO2 component. 

7)    Line 206 “Fig. 5 clearly shows that Ag and Ti elements are distributed in the shell regions.” Ag NPs are “in the shell” or “loaded onto” polymer?

Response: Thanks for your suggestions. Fig. 5 clearly shows that Ag and Ti elements are distributed in the surface regions. Therefore, Ag NPs are loaded onto polymer. In the revised manuscript, we have modified this description. 

8)     Line 221 “Aggregation of the Ag NPs loaded onto … the microgel” No data about the original size of the Ag NPs are presented, in order to claim that they are aggregate on the polymeric particles.

Response: Thanks for your suggestions. With increasing amounts of [Ag(NH3)2]+, the average diameters of Ag NPs loaded onto polymer/TiO2 are approximately 5.8 nm and 10.8 nm, respectively, which are determined by the statistical calculations based on the TEM images (Figure 4). In the revised manuscript, we have added this description.

9)    FTIR spectra of (P(St-NIPAM/P(NIPAM-co-MAA))-TiO2 should be presented and discussed in order to support the conclusions on the interaction of TiO2 with the polymeric chain (see comment on line 89). The spectrum of the polymer-TiO2 composite is missing, thus it is difficult to evaluate the interaction of the polymeric matrix with Ag NPs.

Response: We appreciate for the reviewer’s comments. In the revised manuscript, we have added FT-IR spectrum and the relevant explanation of P(St-NIPAM/P(NIPAM-co-MAA)-TiO2 composite microgels.

The component of the composite microgels was characterized by FT-IR spectroscopy (Figure 7). In the FT-IR spectra of P(St-NIPAM)/P(MAA-co-NIPAM), the peaks at 698 and 757 cm−1 were assigned to the C–H out-of-plane wagging vibration of singly substituted aromatic rings, indicating that styrene is present in the prepared materials. 1649 and 1547 cm-1 were assigned to the C=O stretch of PNIPAM amide (I) and (II) groups, whereas 1709 cm–1 is attributed to the asymmetrical stretching vibration of the C=O group in MAA [45]. In addition, the peaks at 3309 cm-1 and 3440 cm−1 were attributed to N–H stretching vibration of the PNIPAM and the stretching vibration absorption of hydroxyl function groups. Compared with the FT-IR spectra of the template microgels, the peaks at 1649, 1547, and 1709 cm-1 of the P(St-NIPAM)/P(MAA-co-NIPAM)-TiO2 and P(St-NIPAM)/P(MAA-co-NIPAM)-TiO2/Ag composite microgels were weakened to a certain extent, which indicated that a strong interaction occurs at TiO2 and Ag NPs with polymer microgels [Sun, D.; Yang, J.; Wang, X. Bacterial cellulose/TiO2 hybrid nanofibers prepared by the surface hydrolysis method with molecular precision. Nanoscale 2010, 2, 287-292.]. The peaks changing at 3440 cm-1 were attributed to the stretching vibration absorption of hydroxyl function groups (TiO2OH bonds) [Chen, Y.; Xu, X.; Fang, J.; Zhou, G.; Liu, Z.; Wu, S.; Zhu, X.. Synthesis of BiOI-TiO2 composite nanoparticles by microemulsion method and study on their photocatalytic activities. Sci. World J. 2014, 2014, 647040]. FT-IR results showed that there is no characteristic vibration of O-Ti-O bonds in the range of 450-610 cm-1, and TiO2 doesn’t form a complete shell onto P(St-NIPAM)/P(NIPAM-co-MAA) microgels. Thus, the formation TiO2 is mainly present into polymer chains network.

The specific modifications have been denoted in red color in the revised manuscript

10) In order to facilitate the understanding of provided explanation of FTIR data (lines 230-236) the peaks identification in Fig 7 will be useful.

Response: Thanks for your suggestions. In the revised manuscript, we have identified the peaks in FT-IR spectra of the prepared different materials (Figure 7). 

11) Line 278: The loss of mass during the pyrolysis of (P(St-NIPAM/P(NIPAM-co-MAA))-TiO2 and (P(St-NIPAM/P(NIPAM-co-MAA))-TiO2/Ag NPs microgels does not support the statement that “TiO2 and Ag NPs were successfully loaded onto the surface of polymer microgels”. Besides, the TGA analysis is not a relevant method for the characterization of the core-shell morphology and nanoparticles loading in a specific location of a composite material.

Response: We appreciate for the reviewer’s comments. The presence of TiO2 and Ag NPs can be proved by TEM, EDS mapping, XPS and XRD data. In the revised manuscript, we have removed this sentence. 

12) Line 293: What is the relevance of the study of the SPR dependence with the temperature?

Response: We appreciate for the reviewer’s comments. The SPR wavelength of Ag NPs is the relevance with the surrounding microenvironment. For Ag NPs loading thermosensitivity polymer microgels, the SPR wavelength of Ag NPs can be tuned by swelling and shrinkage of polymer with temperature changing. Here, the reversible variation of SPR of Ag NPs with temperature changing is related to the reversible change of refractive index rather than the aggregation[Z. X. Qian, K. N. Guye, D. J. Masiello and D. S. Ginger, J. Phys. Chem. B, 2017, 121, 1092-1099.].

In the revised manuscript, we have added the explanation of the relevance of the study of the SPR dependence with the temperature. The specific modifications have been denoted in red color in the revised manuscript.

13) Line 375: This statement is not supported by data provided- the influence of the TiO2 presence of the Ag NPs size is not studied.

Response: We appreciate for the reviewer’s comments. In this study, we prepared Ag NPs loading onto the P(St-NIPAM)/P(NIPAM-co-MAA) pure polymer microgels and P(St-NIPAM)/P(NIPAM-co-MAA)-TiO2 organic-inorganic composite microgels. Observing the TEM images, it was showed that the sizes and distribution of the formed AgNPs can be well controlled in the presence of TiO2 introduced into the shell-layer network.

14) Line 400: “the AgNPs sizes and loading amounts were clearly 400 influenced by the initial [Ag(NH3)2]+ concentration” No data about the size of Ag NPs is shown in the paper.

Response: We appreciate for the reviewer’s comments. In this study, TEM images (Figure 4) shows that the sizes of the AgNPs and their dispersion are affected by the initial [Ag(NH3)2]+ concentration. Increasing amounts of [Ag(NH3)2]+ led to larger AgNPs sizes and loading amounts. This is ascribable to the enhancement in reduction rate brought about by the higher initial concentration of [Ag(NH3)2]+, leading to the formation of more silver seeds that increasingly grow on the surfaces of P(St-NIPAM)/P(NIPAM-co-MAA)-TiO2 composite microgels. Also, the SPR of the larger AgNPs became red shifted and broadening of the absorption bands with increasing initial concentrations of [Ag(NH3)2]+ (Figure 6). 

15) Line 403 Why to tune the SPR?

Response: We appreciate for the reviewer’s comments. This question is similar to question 12. The SPR wavelength of Ag NPs is the relevance with the surrounding microenvironment.  In the revised manuscript, we have modified the expression of “The SPR of the P(St-NIPAM)/P(NIPAM-co-MAA)-TiO2/Ag composite materials were tuned by changing the environmental temperature” into “The SPR of Ag NPS were tuned by changing the environmental temperature”.

Reviewer 2 Report

The work presented by Zhang et al deals with the synthesis and characterization of Ag-decorated polymer-inorganic composites for uses in catalysis. The work has been carefully conducted and the description of the synthetic and characterization procedures are well described. The materials synthesized are monodispersed polymeric particles that show different swelling properties upon different temperature exposure. The catalytic activity of the new material was described for the reduction of 4-nitrobenzene. Although the experiments were carried out carefully there are some control reactions missing. For instance:

1-    what is the reactivity of the materials in the absence of silver? I.e., P(St-NIPAM)/P(NIPAM-co-MAA)-TiO2 alone? Or the P(St-NIPAM)/P(NIPAM-co-MAA)-TiO2 plus silver ions such as AgNO3? 

2-    How do the authors perform the different catalytic cycles? Do they manage to separate the particles from the solution and reuse the in a new batch?

3-    Is there leaching of the catalytic active sites into the homogeneous phase? 

After the authors address these concerns, I believe this contribution would be ready for publication.

Minor suggestion: could the authors shorten the name of the materials? it is quite distracting and difficult to follow along the lines.

Author Response

The authors appreciate very much for the reviewer’s valuable comments on our manuscript. All the comments have been seriously considered. And following the suggestions, we revised the manuscript with additional experimental data and related discussions provided in the revised manuscript.

Response to reviewer 2:

The authors appreciate very much for the reviewer’s valuable comments on our manuscript. All the comments have been seriously considered. And following the suggestions, we revised the manuscript with additional experimental data and related discussions provided in the revised manuscript.

1) What is the reactivity of the materials in the absence of silver? I.e., P(St-NIPAM)/P(NIPAM-co-MAA)-TiO2 alone? Or the P(St-NIPAM)/P(NIPAM-co-MAA)-TiO2 plus silver ions such as AgNO3

Response: We appreciate for the reviewer’s comments. Addition of P(St-NIPAM)/P(NIPAM-co-MAA)-TiO2, the characteristic UV-Vis absorption peak of 4-NP did not change over 30 min (Figure 13b), indicating that the reduction does not occur in the absence of AgNP.

In the revised manuscript, we have added the data and explanation of the reactivity of the materials in the absence of Ag NPs. The specific modifications have been denoted in red color in the revised manuscript. 

2) How do the authors perform the different catalytic cycles? Do they manage to separate the particles from the solution and reuse the in a new batch?

Response: Thanks for the reviewer’s comments. After each catalytic cycle, the used catalyst was separated from the reaction mixture and washed alternating with double-distilled water and absolute ethanol for several times, and dried in a vacuum oven for 24 h at room temperature. The recovered particles were then used for the next run under the same conditions.

In the revised manuscript, we have added the description for recyclability of the catalysts. 

3) Is there leaching of the catalytic active sites into the homogeneous phase? 

Response: Thanks for the reviewer’s comments. The slight reduction in catalytic activity following recycling is probably the result of the blockage of some active sites in the catalyst and the leaching of active-site Ag NPs into the homogeneous phase.

The specific modifications have been denoted in red color in the revised manuscript. 

Minor suggestion: could the authors shorten the name of the materials? it is quite distracting and difficult to follow along the lines.

Thanks for your suggestions. In the revised manuscript, we have shorten the name of the materials, for example, P(St-NIPAM)/P(NIPAM-co-MAA)-TiO2 named as polymer-TiO2, P(St-NIPAM)/P(NIPAM-co-MAA)-TiO2 /Ag named as polymer-TiO2/Ag.

Round 2

Reviewer 1 Report

The paper could be published in the present form

Reviewer 2 Report

The authors have addressed all the comments from the reviewers. I believe the manuscript is now ready for publication.